# Metrological Protocols for Reaching Reliable and SI-Traceable Size Results for Multi-Modal and Complexly Shaped Reference Nanoparticles

**DOI:** 10.3390/nano13060993

**Published:** 2023-03-09

**Authors:** Nicolas Feltin, Loïc Crouzier, Alexandra Delvallée, Francesco Pellegrino, Valter Maurino, Dorota Bartczak, Heidi Goenaga-Infante, Olivier Taché, Sylvie Marguet, Fabienne Testard, Sébastien Artous, François Saint-Antonin, Christoph Salzmann, Jérôme Deumer, Christian Gollwitzer, Richard Koops, Noham Sebaïhi, Richard Fontanges, Matthias Neuwirth, Detlef Bergmann, Dorothee Hüser, Tobias Klein, Vasile-Dan Hodoroaba

**Affiliations:** 1Laboratoire National de Métrologie et d’Essais (LNE), 29 Avenue Roger Hennequin, 78197 Trappes, France; 2Dipartimento di Chimica and NIS Inter-Department Centre, University of Torino, Via P. Giuria, 10125 Torino, Italy; 3National Measurement Laboratory, LGC Limited, Queens Road, Teddington TW11 0LY, UK; 4CEA, CNRS, NIMBE, Université Paris-Saclay, 91191 Gif-sur-Yvette, France; 5CEA, Liten, DTNM, Université Grenoble Alpes, 38000 Grenoble, France; 6Federal Institute for Materials Research and Testing (BAM), Unter den Eichen 44-46, 12203 Berlin, Germany; 7Physikalisch-Technische Bundesanstalt (PTB), Abbestraße 2–12, 10587 Berlin, Germany; 8VSL National Metrology Institute, Thjsseweg 11, 2629 JA Delft, The Netherlands; 9National Standards (SMD), FPS Economy, 16 Bd du Roi Albert II, B-1000 Brussels, Belgium; 10Pollen Metrology, 122 Rue du Rocher de Lorzier, Novespace A, 38430 Moirans, France; 11Physikalisch-Technische Bundesanstalt (PTB), Bundesallee 100, 38116 Braunschweig, Germany

**Keywords:** certified reference nanomaterials, traceable nanoparticle size measurements, hybrid metrology, scanning probe microscopy, small-angle X-ray scattering, electron microscopy

## Abstract

The study described in this paper was conducted in the framework of the European nPSize project (EMPIR program) with the main objective of proposing new reference certified nanomaterials for the market in order to improve the reliability and traceability of nanoparticle size measurements. For this purpose, bimodal populations as well as complexly shaped nanoparticles (bipyramids, cubes, and rods) were synthesized. An inter-laboratory comparison was organized for comparing the size measurements of the selected nanoparticle samples performed with electron microscopy (TEM, SEM, and TSEM), scanning probe microscopy (AFM), or small-angle X-ray scattering (SAXS). The results demonstrate good consistency of the measured size by the different techniques in cases where special care was taken for sample preparation, instrument calibration, and the clear definition of the measurand. For each characterization method, the calibration process is described and a semi-quantitative table grouping the main error sources is proposed for estimating the uncertainties associated with the measurements. Regarding microscopy-based techniques applied to complexly shaped nanoparticles, data dispersion can be observed when the size measurements are affected by the orientation of the nanoparticles on the substrate. For the most complex materials, hybrid approaches combining several complementary techniques were tested, with the outcome being that the reliability of the size results was improved.

## 1. Introduction

In the framework of the European project nPSize (EMPIR program), six national metrology and designated institutes (BAM, LGC, PTB, LNE, SMD, and VSL), the French Research Centre CEA, the University of Turin (UNITO), and a French SME (Pollen Metrology) gathered together with the main aim of improving measurement capabilities for the determination of the dimensional properties of various nanoparticle (NP) populations based on both measuring methods traceable to SI units and new reference materials. Firstly, the performance of several NP characterization methods was evaluated in terms of their sensitivity to material (chemical nature), shape, and polydispersity for representative nanoparticulate materials (i.e., metals and oxides). A set of validated nanoparticle reference materials was synthesized with (i) non-spherical shapes, (ii) non-monodisperse size distributions, and (iii) accurate concentrations. In order to improve the evaluation of the nanoparticle measurement uncertainty and comparability between the results of different methods, improved physical models of the output signals obtained during the data acquisition process were developed. Finally, the results of this project have made a significant contribution to the standards development work of the technical committees CEN/TC 352 Nanotechnologies and ISO/TC 229 Nanotechnologies.

Several techniques were selected for their proven performance and their ability to provide reliable results in NP size measurement: three electron microscopy-based (EM) techniques (transmission electron microscopy—TEM, scanning electron microscopy—SEM, and transmission scanning electron microscopy—TSEM), a scanning probe microscopy technique (atomic force microscopy—AFM), and an ensemble method, small-angle X-ray scattering (SAXS), capable of measuring the size of NPs in powder or in suspension. Microscopy techniques are counting methods and give local information on single NPs. The size is deduced by counting NPs one-by-one in one or more images and after building a histogram of the size distribution. The measured NPs in the histogram must be statistically representative of the whole sample population. Sample preparation is therefore of crucial importance. Deposition protocols have been developed to obtain NPs homogeneously dispersed and isolated on flat substrates in order to accurately determine the NPs’ size distribution. SAXS is an ensemble method that measures the differential X-ray scattering cross-section of a sample, i.e., the sum of the scattering signals of all NPs present in the sample as a function of the scattering angle. Fitting the total signal with an appropriate model function allows the extraction of the parameters of the underlying model, including the assumed size distribution [1].

In the nPSize project, the focus has been set on: (i) the size traceability chain for complex shaped (non-spherical) and polydisperse NP populations, (ii) clear measurand definitions for an optimized determination of characteristic parameters for each NP geometry, and iii) the improvement of the measurement process steps. Regarding microscopy techniques, it is generally accepted that the size analysis process consists of four different steps [2,3]:iSample preparation: The protocol must be optimized to minimize additional NP agglomeration. The presence of well-dispersed homogeneous isolated NPs on the substrate facilitates accurate and reliable NP size results. Moreover, automated analysis is thus enabled.iiCalibration and metrological qualification of the instrument: Calibrating the instrument means creating a correlation between the obtained measurement result and the definition of the unit concerned in the SI (Système International). This periodically performed action makes the measurement result reliable and comparable with other laboratories with the same or other techniques. The calibration process can be carried out with certified reference materials. The metrological qualification of the instrument allows the operator to identify all error sources affecting the measurement result and hence quantitatively assess an uncertainty budget associated with the measurement.iiiData acquisition: The term “measurand” is linked to the quantity to be measured and must match the dimensional descriptors (particle shape and particle size) [2] of the nano-object to be determined as accurately as possible, describing its geometry as completely as possible. For this study, the near-spherical NPs can be defined by a single parameter, but the more complex geometries will need at least two parameters. Ideally, the measurand’s definition must be as specific as possible including the terms: mean, median, or mode. In this paper, all the reported results are the mean values of the respective measurands.ivMicroscopy image and data analysis process: This is a key step because the size measurand value is extracted from the image through software provided with mathematical tools. Several software packages suitable for NP size exist on the market (e.g., MoutainsMap^®^ and Digital Surf) or as open access software (e.g., ImageJ and Gwyddion). However, algorithms determining the NP size from the EM raw signal are considered as a black box and the common “watershed” algorithms capable of identifying the NP boundaries within agglomerates are often unsatisfactory for reliable segmentation. That is the reason why we have decided for the microscopy techniques in this study to handle measurements only on isolated NPs. Furthermore, image analysis software, Platypus ^®^, was specially designed by Pollen Metrology for the nPSize project and adapted for EM and AFM images.

As part of the continuous improvement of the traceability chains regarding the dimensional measurements of particles at the nanoscale, there is great interest in identifying and developing new candidates of certified reference materials. This applies to all types of instruments and measurement methods, but in particular, microscopy-based techniques are targeted. For that purpose, nPSize has developed and validated three classes of candidate reference (test) materials (RTM): (i) with a well-defined non-spherical shape, (ii) of relatively high polydispersity index, and (iii) of accurate particle concentration. In this paper, we report on two different types of nanomaterials: (i) three samples of NPs with various complex shapes and (ii) two (gold and silica) bimodal populations of NPs considered as spherically shaped.

The approach used for harmonizing the measurement process for each characterization method was the organization of an inter-technique and inter-laboratory comparison (ILC). In the past, numerous ILC exercises have already been conducted for determining the NP size distribution, but they were almost exclusively dedicated to spherical, monomodal, monodispersed, and easily dispersed particles. The investigated samples were often certified reference materials of metals (gold and silver), metal oxides (SiO_2_), or perfectly spherical polymers [4]. The best results (σ_interlab._ < 1 nm, data dispersion among laboratories) were obtained on gold nanoparticles when the measurements were performed with a single electron microscopy technique (TEM) by using automated particle analysis [5]. Some studies involved several direct and indirect methods based on different physical principles. DLS (dynamic light scattering) and DMA (Differential mobility analyzer) measurements were compared with the microscopy results (AFM and SEM) obtained on polystyrene and silver NPs with data dispersion larger than 5 nm [6]. SAXS was added to DLS, CLS (centrifugal liquid sedimentation), and electron microscopy techniques (SEM and TEM) in an ILC exercise using silica NPs showing consistent data with a standard deviation of 2.5 nm [7]. Two comprehensive studies were conducted by specifying the calibration processes, uncertainty evaluations, some details on the measurands, and involving indirect (DLS, DMA, XRD, and SAXS) and direct (TSEM, SEM, AFM, and TEM) methods [8,9]. As per the investigated samples (gold, silver, and polystyrene), the results are disparate, and the standard deviations among the techniques can reach 6 nm for spherical and monomodal populations when only SAXS and microscopy measurements are considered. Finally, more recently, several papers have dealt with measurement comparisons performed with microscopy techniques (TEM and AFM) on more complexly shaped particles (carbon black aggregates, nanorods, cellulose, and titania) [10,11,12,13,14].

In the present study, each sample was circulated among the project partners and the precisely defined measurands were measured according to protocols discussed and agreed in advance. The microscopy laboratories (AFM, SEM, TEM, and TSEM) implemented their own protocols of sample preparation specifically developed to reach an optimized dispersion of NPs on the substrate and most partners used Pollen’s software for analyzing the images and determining the size of the NP populations. The sizes obtained with SAXS, as an ensemble method, were compared with those given by microscopy, which is a local and counting method. Specific algorithms have been developed in nPSize to model complex shapes of NPs such as bipyramids (nPSize03) and nanocubes (nPSize04) [1].

## 2. Materials and Methods

### 2.1. Material Preparation

Eleven new different types of materials were prepared in the nPSize project for testing purposes. In this paper, only five have been selected for systematic characterization, see details in Table 1. Multi-modal nearly spherical as well as complex-shaped (cubes, bipyramids, and rods) NPs were chosen for testing the performance of the microscopy techniques and the consistency with SAXS regarding their ability to measure the NP size distribution accurately.

More details about the synthesis of the samples are given in Appendix A.

### 2.2. Measurement Techniques

#### 2.2.1. Transmission Electron Microscopy (TEM)

TEM analyses were carried out using a TECNAI OSIRIS (Thermo Fisher Scientific, Eindhoven, Netherlands) transmission electron microscope, equipped with a high sensitivity GATAN camera (BM-Ultra-scan, Pleasanton, CA, USA), with 2048 × 2048 pixels. The acceleration voltage was 200 kV using the machine-specific “index 5” for the spot size to reduce possible sample damage induced by the high beam current. An adjustment of the magnification was made to obtain a good pixel number per particle and the number of particles needed for good statistics corresponding to a number of images between 15 and 25. A TIFF image format was used, derived from the GATAN proprietary format (DM4). Here, the method used to determine the dimensional properties of a population of NPs by TEM is based on the thresholding tool developed in the frame of the project by Pollen and included in the Platypus™ software (version specifically developed for nPSize project, 2018, Pollen Metrology, Grenoble, France). At least 100 particles were thresholded and the distributions of the minimum Feret, maximum Feret, and equivalent circle diameters were extracted by using the software.

#### 2.2.2. Scanning Electron Microscopy (SEM)

Two laboratories carried out the NP measurements with SEM. The first instrument (SEM-1) is a Zeiss-Ultra+ field-emission gun (FEG) microscope equipped with a GEMINI column and an in-lens secondary electron detector (Zeiss, Oberkochen, Germany) particularly useful for measuring nano-objects due to its high sensitivity to the surface morphology. The Zeiss-Ultra+ resolution is specified by the manufacturer as 1.7 nm at 1 kV and 1.0 nm at 15 kV. The second instrument (SEM-2) is also a Zeiss microscope of type Supra 40 equipped with a Schottky field emitter (Zeiss, Oberkochen, Germany). Regarding both SEMs, the measurements were performed with the secondary electron in-lens detector.

#### 2.2.3. Scanning Electron Microscopy in Transmission Mode (TSEM)

In SEM, not only the secondary electrons but also the transmitted electrons may be used for image formation, with them having a number of advantages such as improved resolution and contrast [22] ultimately leading to easier and more accurate discrimination of the imaged particles from the background [23,24]. One should note that there are different denominations for the transmission mode in SEM, which corresponds to the STEM mode of a TEM, but with acceleration voltages of a maximum of 30 kV: STEM-in-SEM, TSEM, T-SEM, LV-STEM, etc. For uniformity reasons, we call it “TSEM” in this paper. Details on the various transmission approaches used in this study by different laboratories (either in the configuration with a STEM detector or by using a dedicated transmission sample holder) are given in Ref [23,24]

TSEM-1 and TSEM-2 denote two instruments with dedicated transmission detectors: Zeiss Supra 35 VP (Zeiss, Oberkochen, Germany) and Thermo Fisher Verios G4 (Waltham, MA, USA), respectively. For imaging, the so-called bright field detector is used which detects electrons that transmit through the sample without significant deflection. For the subsequent image analysis and sizing, homemade software is used, which considers an iterative procedure to separate the particles from the background in an image and which is based on Monte Carlo simulations of the image formation process [22]. For spherical particles measured with a Zeiss Supra 35 VP the procedure is detailed in ref. [22]. Micrographs of non-spherical particles are taken with a Thermo Fisher Verios G4 and subsequently analyzed using an adapted approach also based on simulations.

A third approach (TSEM-3) uses a dedicated transmission sample holder which converts the transmitted electrons into secondary electrons that are collected by the Everhart-Thornley detector (Zeiss, Oberkochen, Germany) [23,25]. This version has been used in conjunction with a Zeiss SEM of type Supra 40 equipped with a Schottky field emitter, which also produces bright field images.

In all three TSEM configurations, samples prepared on conventional TEM grids are used.

#### 2.2.4. Atomic Force Microscopy (AFM)

Three different AFM instruments were used in this study: (i) an Oxford Instrument Asylum Research MFP3D Infinity (AFM-1) (Oxford Instrument Asylum Research, Santa Barbara, CA, USA) with a maximum scan field of 90 µm × 90 µm and a height range of 15 µm; all the measurements were carried out in tapping mode using Nanosensors PPP-NCHR probes (Nanosensors, Neuchatel, Switzerland) (about 7 nm nominal radius of tip apex; the images were recorded with a pixel size of 5 nm and a scan speed between 4 and 7 µm/s; (ii) a Veeco Dimension 3100 (AFM-2) (Bruker, MA, USA) with a maximum scan field of 100 µm × 100 µm and a height range of 5 µm. Measurements were acquired in tapping mode with a constant scan speed of 3 µm/s irrespective of the size of the scan field. Various probe types were used depending on the samples; and (iii) a Veeco Dimension 3100 with Nanoman V controller (AFM-3) (Bruker, MA, USA) with an accurate three-axis scanner operating under closed-loop control (hybrid XYZ-scanner with a range of 90 µm × 90 µm × 8 µm). OTESPA-R3 probes (about 7 nm nominal radius of apex) (Bruker, MA, USA) were mounted, and the tapping mode was used for all samples. All of the images were recorded with a pixel size of 5.0 nm. The scan speed was maintained at 4 µm/s with constant feedback parameters.

The AFM images (AFM-2, AFM-3) were processed using Platypus software (version specifically developed for nPSize project, 2018, Pollen Metrology, Grenoble, France) except for images acquired with AFM-1 analyzed with the Image Metrology SPIP software (6.7.5 version, 2018, Image Metrology A/S, Lyngby, Denmark).

#### 2.2.5. Small Angle X-ray Scattering (SAXS)

The SAXS experiments were conducted at the four-crystal monochromator beamline of the PTB laboratory at the synchrotron radiation facility BESSY II at a photon energy of 8 keV [26]. All measurements were performed with NPs in suspension, confined in capillaries with rectangular cross-sections made of borosilicate glass, which were mounted vertically in the vacuum sample chamber. These capillaries have (Hilgenberg GmbH, Germany) of 80 ± 0.5 mm, an outer width of 4.2 ± 0.2 mm, an outer thickness of 1.25 ± 0.05 mm, and a single wall thickness of 120 µm. These capillaries have a high degree of homogeneity with respect to their thickness along their vertical axis and were vacuum-sealed for the experiment. To obtain better statistics, we measured at different points along the vertical axis of the capillary and averaged the corresponding scattering curves to obtain an overall curve that was eventually evaluated. In this way, we can also detect possible sedimentation of the particles within the capillary. In addition to the sample, Fluorinert F-3283 was filled into the bottom of the capillary, this fluid does not mix with the NP aqueous suspension filled on top of it. The optical path through the sample thickness was determined from the transmission through the Fluorinert layer with a known attenuation coefficient [27]. For the fitting of the averaged SAXS curve, a model must be used, e.g., Debye’s scattering equation [1].

### 2.3. Sample Preparation for Microscopy-Based Techniques

#### 2.3.1. Deposition on the Copper Grid (TEM and TSEM)

For TEM, TSEM, and possibly also SEM examination, the nanoparticles that are delivered in suspension were deposited on carbon TEM grids. While the details of the preparation procedure vary between institutes and possibly also between samples, they all consist of the following five steps: optional pre-treatment of the TEM grid, agitating the vial, deposition of a droplet of suspension, incubation, and optional rinsing.

Prior to the preparation, the hydrophilicity of the TEM grid may be enhanced by ozone or plasma treatment. Furthermore, its adhesion properties may be improved using poly-L-Lysine. Before the deposition, the vial that holds the nanoparticle suspension is agitated either by hand, by a shaker, or using an ultrasonic bath. Thereafter, a droplet of the suspension is deposited on the TEM grid using pipettes, syringes, or the like.

The simplest incubation approach is drying the drop on air or protective gas, but if there are issues with drying artifacts, either the incubation time may be reduced or incubation takes place in a humid environment. In these cases, the excess liquid must be removed simply by using blotting paper or by more sophisticated techniques such as spin coating, which also helps to achieve an even distribution of nanoparticles on the substrate.

After an optional rinsing step, the samples are then ready for examination in the electron microscope.

#### 2.3.2. Deposition onto Mica or a Silicon Wafer (SEM and AFM)

Three different substrates were used for SEM and AFM measurements: (i) mica (AFM-1 and 2), (ii) silicon wafer (SEM-1 and AFM-3), and (iii) aluminium SEM stub (SEM-2). All these substrates were preliminarily functionalized with PLL to improve NP–substrate adhesion and prepared following the protocol described in Section 2.3.1. Since the surface area of these substrates is noticeably higher than that of the used TEM grid, the deposited volume of PLL is adapted to 50 µL.

The mica is particularly suitable for sensitive height measurements by AFM (AFM-1 and AFM-2) because the surface roughness is very low [28]. The mica substrate must be cleaved, for instance, by pressing a piece of adhesive tape against the surface of the mica disc and then smoothly removing the tape. The top layer of the mica should stick to the tape. This must be repeated several times until a full layer is removed and the exposed surface is smooth. Fifty µL of NP suspension is required for covering the whole surface area of the mica disc of roughly 10 mm diameter. The suspension shall incubate for at least 30 min; the substrate is then rinsed with distilled water and dried by blowing clean air. All the NP samples were deposited on PLL-modified mica substrates following the described protocol except for the gold nanorods (nPSize07), that were deposited directly on the cleaved mica substrate, without PLL. Mica is a poor electrical conductor and particles deposited on this type of substrate can be damaged when analyzing them by SEM. It is therefore preferable to use silicon wafers for instance.

The protocol of NP deposition onto the silicon substrate (SEM-1 and AFM-3) is slightly different and relies on a spin-coater. A 7.5 µL droplet of NP suspension is deposited on the center of a silicon wafer fixed on the sample holder of the spin-coater. A first step at a low spin speed (1000 rpm) for a 60 s time interval allows the suspension to spread over the substrate surface. The rotation time controls the surface density of NPs. A second step at 8000 rpm for 10 s induces rapid drying. This protocol prevents the NP agglomeration phenomenon occuring [28].

## 3. Instrument Calibration and Measurement Procedures

### 3.1. TEM Measurement Traceability

A cross-grating sample (“Calibration grating Replica” from Ted Pella (Ted Pella, CA, USA) was used at different magnifications following the calibration procedure defined and integrated into the software by the instrument manufacturer. The main contributions of measurement uncertainty sources are detailed in [29] with a dedicated assessment of the bias induced by the out-of-focus position when using the sample nPSize01 (gold NPs). The main contributions to the uncertainty budget are reported in Table 2.

### 3.2. SEM Measurement Traceability

The calibration process of SEM-1 begins with the image pixel size calibration against a specific transfer standard (*P*900*H*60, CNRS/C2N, Palaiseau, France) by measuring the pitch in X and Y directions using Fourier transform [28]. P900H60 was calibrated against metrological AFM (*mAFM*) beforehand, and a correction factor can be determined:(1)$$a_{P900H60}=\frac{d_{SEM}}{d_{mAFM}}$$
with *d_SEM_* and *d_mAFM_* denoting the pitch measurement carried out with *SEM* and *AFM*, respectively.

In a previous paper, an uncertainty budget associated with SEM-1 had been established for the mean diameter measurements of reference silica NPs [28]. The main uncertainty sources are summarized in Table 3. Sample preparation is a key step of the measurement process and can have a significant impact on the uncertainty assessment. In the case of a protocol minimizing the agglomeration phenomenon, this impact can be noticeably reduced. The thresholding is also an operator effect potentially with a major contribution to the uncertainty budget. Finally, concerning contamination, pumping the SEM chamber with the sample for 24 h prevents covering the nanoparticles with carbon layers when the e-beam is applied. Without this caution, the contamination disturbs the measurements.

### 3.3. TSEM Measurement Traceability

The traceability of TSEM measurements relies on the use of 2D grating with a nominal grating pitch of 144 nm of aluminum bumps on silica (150-2D from Advanced Surface 150-2D., Indianapolis, IN, USA) and a laser diffractometer which yields traceable values for the mean grating pitch. The traceability to the meter is realized at the primary level in terms of the wavelength from a helium-neon laser. The diffractometer is used to determine the pitch, *a*_optical_, of the 2D grating. The latter is also measured by TSEM and *a*_TSEM_ is determined. The pixel size, α_pixel_, is then calculated as a traceable result:α_pixel_ = *a*_optical_/*a*_TSEM_(2)

There are numerous contributions to the measurement uncertainty, which are roughly classified into five groups to foster a brief discussion, see Table 4. As described above, the pixel size can be determined quite accurately and thus it hardly contributes to the overall uncertainty. TSEM images show a pixelated 2D projection of 3D particles which could possibly be altered by the electron beam, which altogether leads to a medium uncertainty contribution. Sample preparation should ensure that a representative fraction of preferably isolated particles is evenly distributed across the TEM grid, a process that is mostly afflicted with significant uncertainties. Only a fraction of the particles on the TEM grid are imaged and not all of these are analyzed, leading to a medium uncertainty contribution. The largest uncertainty contribution comes from the need to separate background pixels from pixels belonging to the particle—the so-called thresholding, which may be supported by Monte Carlo simulations. Its impact obviously depends on the pixel size and becomes more significant for more complex particle shapes. Thus, it is highest for the bipyramides with ambiguous geometry and undefined orientation.

### 3.4. AFM Measurement Traceability

Along the Z-axis, the calibration procedures for AFM-1, AFM-2, and AFM-3 were carried out either (i) by comparison to physical step height standards via a calibration curve (for h > 20 nm), (ii) through a virtual height standard (for h < 20 nm), or (iii) by comparison to a reference structure specifically developed and measured by the mAFM.

In the first case, the calibration curve of the microscope (AFM-1) is determined by fitting the standard reference mean step height values of several step height standards as a function of the measured mean step height values. Then, the spherical NP dimensions are measured and related to the calibration curve. The traceability of these physical step height standards is realized by interference microscopy or using an mAFM directly traceable to the SI. The traceability method and uncertainty budget evaluation for AFM-1 are detailed in ref. [30].

In the second case, a calibrated piezo-actuator applies a known periodic displacement to the virtual standard or selects an appropriate step height value close to the nominal height of the particles to be measured (AFM-2). The values generated by the virtual standard are traceable by laser interferometry and the linearity of the displacement versus square wave amplitude is nearly perfect in the range from 2 pm to 20 nm. The calibration of the AFM-3 along the Z-axis was carried out by using a specific reference structure (P900H60, CNRS/C2N, France), preliminarily measured by the mAFM. The process is detailed in ref. [28]. The identified uncertainty sources for AFM height measurements of the NP materials used in this project are the uncertainty of the calibration of the physical and virtual height standards, the tapping force through the amplitude setpoint, the scan speed, the operator, the image analysis, the repeatability of the measurements, and the size dispersion of the particles, see Table 5.

### 3.5. SAXS Measurement Traceability

A properly normalized scattering curve shows the differential scattering cross-section per volume of sample (absolute scattering intensity) as a function of the momentum transfer *q* of the X-ray photons. The correct *q*-scaling is calculated from a known sample–detector distance, the beam center position on the detector, as well as the photon energy. For calibration of the intensity, the ratio between the intensity of the scattered photons and that of the incident photons must be calculated. This can be achieved by calibration of the detector in use, a measurement of the direct beam with the scattering detector, or calibration with scattering standards such as water or glassy carbon. Additionally, the length of the optical path through the sample (thickness) must be known. For isotropic particle orientations, the detected 2D scattering signal consists of concentric circles such that azimuthal integration and proper normalization (by incident photon flux, exposure time, the solid angle of the scattered beam, the quantum efficiency of the detector, sample transmission, and sample thickness) of the photon counts lead to a 1D scattering curve. This curve can then be fitted by an appropriate model function which depends on the particle shape. More details on the calibration, model fitting, and physical background can be found in ref. [31]. Table 6 gives the main uncertainty contributions impacting the NP size measurements by SAXS.

### 3.6. Measurands and Descriptors

By definition, the measurand corresponds to the quantity to be measured. Table 7 displays the various parameters required for defining the shape of the studied NPs as well as the measurands associated with each technique. The spheroidal NPs (nPSize01 and nPSize12) and the cubic NPs (nPSize04) are properly described by one single parameter. The equivalent circular diameter (ECD), *D*_eq_, is regularly used in electron-microscopy-based (EM) images and corresponds exactly to the physical diameter in the case of a perfect sphere. The ECD of a nano-object represents the diameter of a circle that occupies the same two-dimensional surface area as the object. In this study, only the NP height is measured by AFM (i.e., the maximum height of the particle relative to the mean height of the substrate background) because the tip/sample convolution effects significantly impacts the NP lateral dimension measurements. This height should be equal to *D_eq_* when the NP is perfectly spherical. *D*_SAXS_ is determined from a model taking into account a spherical shape for nPSize1 and nPSize12. Regarding the cubic shape, the particle cube side is measured by *D*_MinFeret_. In order to include the potential irregularities as part of the real particle geometry, *D*_max.⟂_ is measured as well. It should be noted that here we take an alternative definition of the maximum Feret diameter, denoted as *D*_max.⟂_. Formally, *D*_max.⟂_ corresponds to the longest dimension of the particle whatever its orientation is [2]; in our study, it represents the distance between two parallel tangential lines in the direction perpendicular to the *D*_MinFeret_. In other words, according to the standardized terminology, *D*_max.⟂_ for a nanocube is equal to the diagonal length of the square face, whereas, in this paper, it corresponds to the second adjacent side.

To obtain a fair description of the two elongated materials, the bipyramidal NPs (nPSize03) and the nanorods (nPSize07), two parameters are necessary: the length of both structures and the side of the square base and the section, respectively. For determining the nano-bipyramid parameters by SAXS, a specific model integrating a comprehensive description of the geometry was used [32].

## 4. Results and Discussion

### 4.1. Near-Spheroidal Nanoparticle Bimodal Populations

A bimodal population of gold NPs was measured by EM, AFM, and SAXS. Two laboratories have used both imaging techniques, SEM and AFM. The nPSize01 material was considered as consisting of near-spherical particles. For this reason, the measurands *D*_eq_ and *h*_AFM_ were found to be adequate to determine the particle diameters. An overview of the obtained results associated with error bars is given in the graph in Figure 1. The mean sizes of the two modes were found to be at (61 ± 2.3) nm and (31 ± 1.8) nm (see the dotted lines in Figure 1), considering to a normal distribution for each size mode. These values were calculated as the mean and standard deviation of the results obtained by all laboratories. Good consistency is observed among the results for both modes between all techniques and different laboratories, with a dispersion of data below 1 nm for the EM-based techniques. If we add the AFM measurements, the deviation increases roughly to 2 nm for mode 2. Hence, the results are compatible with the measurement uncertainties. Furthermore, the NP sizes obtained by SAXS are very close to the microscopy results.

It should be noted that this kind of NP material is very easily dispersed on the substrate and the agglomeration phenomenon is practically inexistent (see Figure 2). However, the SEM images show that the shape of the nPSize01 is not necessarily a perfect sphere, with extreme cases of deviations from the spherical shape, clearly visible (polyhedra). Thus, we questioned the impact of this large shape polydispersity on the accuracy of the measurements. According to Ref. [33], three different structural shapes can be observed among the gold colloidal crystals. The structure of each particle depends on the growth mechanism: homogeneous (direct NP or cluster aggregation) and heterogeneous aggregation. These various mechanisms can result in amorphous particles, single-crystalline, or polycrystalline clusters. Short-range interactions between particles induce the formation of amorphous NPs with a spherical shape (Figure 2a,b). Single-crystalline NPs are spherical for the smallest among them, and further growth induces the formation of the truncated tetrahedral shape (Figure 2e,f). The polycrystalline structures correspond to icosahedral and decahedral shapes and are terminated by (111) planes (Figure 2c,d). A detailed inspection of the images shows that the population with the smallest mode (31 nm) is noticeably more spherical. The second mode (61 nm) consists of particles with visible faces related to (111) and (100) crystal planes. The presence of triangular particles (<2%), deviating considerably from the spherical shape, and more generally the shape polydispersity observed in this nanogold population do not affect the consistency of the results obtained with the different techniques.

The second material consisting of silica NPs, i.e., nPSize12, was synthesized by the Stöber’s method [16] and is expected to be significantly more spherical than nPSize01. A representative SEM image is given in Figure 3 and shows clearly that the sample consists of two spherical NP populations with two different sizes both at the nanoscale. Even if the dispersion state on the substrate is similar to the sample nPSize01, the presence of a significant number of dimers and even a few trimers is observed, mainly regarding the smaller population. Some dimers seem to originate from the merging of two NPs (simply touching particles or growning together, see Figure 3, right, insert), and their contours at their joint side are difficult to distinguish—at least based only on these SEM images. Nevertheless, independently of the sample preparation protocol used for the microscopy techniques, the NPs are uniformly and properly dispersed on the substrate for achieving optimal measurement results.

The nPSize12 material was measured by electron-microscopy-based techniques (TEM, TSEM, SEM), AFM, and SAXS. The results of the techniques involved are overall very consistent with a data dispersion of 1 nm and 2 nm for the (58.0 ± 1.0) nm and (29.8 ± 1.6) nm modes, respectively. Figure 4 groups all the measurements with associated error bars (k = 2). These values were calculated as the mean of the results obtained by all laboratories. The size distribution of each mode is lognormal. The spread of data is slightly wider for the 30 nm mode due to the presence of merged dimers, which were either taken into account or not in the size distribution. Nevertheless, we can conclude that the presence of dimers and trimers has an insignificant impact on the measurement results.

### 4.2. Bipyramidal Titania Nanoparticles

A series of samples of nano-sized bipyramid particles (nPSize03, nanobipys) were synthesized by the University of Turin. Figure 5 shows representative SEM images of the nPSize03 material after applying the corresponding deposition protocol. Some agglomerates were observed; however, the particles are fairly well-dispersed over the substrate for an optimal measurement process. These NPs are (anatase) truncated tetragonal bipyramids with eight equivalent {101} faces, four {100} faces on the edge of the square base, and two {001} upper faces [32]. According to Figure 5, two parameters must be determined for the accurate description of this complex shape: *L*, the major axis length and *s*, the side of the square base.

The results of the size measurements are reported in Figure 6. Regarding the EM techniques (TEM, TSEM, and SEM), the parameters *s* and *L* can be theoretically determined through the measurands *D*_MinFeret_ and *D*_MaxFeret_, respectively (see Figure 6). The AFM heights, *h*_AFM_, allow for determination of the side of the square base and the obtained values are consistent with *D*_MinFeret_ from the SEM measurements, which was expected. The parameter *s* has been found to scatter between 40 nm and 48 nm, and the *L* measurement results are similarly dispersed (59–69 nm); however, they are associated with larger uncertainties.

The discrepancies between the techniques can be explained by the different orientations of the bipyramids on the substrate. As a matter of fact, regarding the sample preparation for microscopy measurements, a previous study demonstrated that the isolated nanobipyramides could be positioned basically in two ways on the substrate [32]. In orientation 1, one {101} face of the particle is in contact with the substrate, and the second orientation (2) corresponds to the bipyramid balanced on one {100} face of the edge of the square base (see Figure 7). If NPs are positioned according to 2, the measurements of the *D*_MinFeret_ and *D*_max.⟂_ allow both parameters, *L* and *s*, to be directly determined. In contrast, in the case of orientation 1, only the parameter *s* can be directly addressed by SEM through the minimum Feret diameter. The length of the nanobipyramides is difficult to measure due to the non-zero inclination of the major axis to the substrate.

Two thousand NPs were imaged by SEM and the histograms of size distribution related to *D*_MinFeret_ and *D*_max.⟂_ measurands are given in Figure 8. The *D*_max.⟂_ histogram is clearly the convolution of two lognormal distributions (20:80% area ratio), with a mode very close to the one of the *D*_MinFeret_ histogram for the smaller particles. Hence, one can conclude that the nPSize03 material likely consists of two populations with different aspect ratios (AR), as shown in Figure 8 (bottom). The NPs with a roughly 0.75 AR are completely formed, have finished growing further with a larger size, and have a real bipyramid shape. The second population with an AR close to 1 is instead composed of (nearly) spheroidal particles with a smaller dimension and hardly discernible faces.

The study detailed in [32] allowed us to discriminate the nanobipys oriented 1 and 2 by two approaches: (i) using AFM on a population of 250 NPs and (ii) measuring the true value of the length of the “oriented as 1” nanobipyramides major axis by tilting the SEM stage. The results of the *s* and *L* measurements obtained for both orientations are reported in Figure 9. In the same figure, the Feret diameter mean values of the two populations observed in the histograms of Figure 9 as well as the SAXS measuring results are added. From the SEM data, we can conclude that the biggest well-formed nano-scaled bipyramids are statistically more positioned according to orientation 1, but when the AR increases towards close to 1, orientation 2 becomes the most likely one. To corroborate the results obtained for both parameters *s* and *L*, SAXS measurements were carried out for the sample in liquid suspension. A model specifically developed in the framework of the nPSize project and well adapted to bipyramidal shape was used for determining *s* and *L*. Unlike microscopy techniques, SAXS is an *ensemble* method [34] with the final result being independent of the orientation of the nanobipys in the sample. This is because the nanoparticles are suspended and isotropically aligned. Microscopy techniques are used as counting methods and give nanoscale local information. This represents a significant advantage of the microscopical techniques but raises the issue of how statistically representative the EM and AFM results are. Note that Figure 9 demonstrates very good consistency between the SAXS and microscopy results. The value of the parameter *L* is deduced indirectly by first determining the parameter *s* through the fit for which the simulated scattering curve is more sensitive with respect to a shift on the q-axis. Then, *L* can be calculated from the known but fixed AR of the particle model used [32].

The value of the parameter *L* obtained by SAXS is exactly ranged between the major axis lengths as measured by SEM for the particles oriented 1 and 2. In the case of complex-shape NP populations, a combination of EM and SAXS techniques might be a suitable approach for a reliable method to determine the characteristic dimensional parameters. A quick, “descriptive” EM image could give useful information about the NP morphology so that geometry data can then be integrated into the SAXS model for an ensemble analysis.

### 4.3. Gold Nanocubes

The cubic shape of this NP population (nPSize04) seems to be less complex than nanobipyramids, because only one parameter is (ideally) required for describing the entire shape. According to Figure 10, the measurement of the side, *s*, is sufficient, and in the case of a perfect cube, the values of *D*_MinFeret_, *D*_max.⟂_, and *h*_AFM_ should be theoretically identical. Figure 10 displays the results of the size measurements associated with error bars (k = 2) carried out by TEM, TSEM, SEM and AFM, which are very consistent. A fairly low discrepancy of (3.4 ± 0.9) nm between *D*_MinFeret_ and *D*_max.⟂_ values obtained by electron microscopy techniques (TEM, TSEM, and SEM associated with a standard deviation) was measured, which is quite close to the estimated uncertainties. However, the nanocube height found by AFM was 6.3 nm lower than the mean lateral dimensions by EM. However, this result is based on a unique series of measurements performed by only one laboratory. The large uncertainty (5 nm) associated with AFM measurements is explained by the insufficient quality of the prepared sample and the small number of measured particles.

The SAXS measurements show a discrepancy with the EM results but are consistent with the AFM heights. Figure 11 reports the experimental SAXS curve of the nPSize04 material.

A detailed description and discussion of Figure 11 can be found in Ref [1]. For example, the authors also show that a spherical model function would not fit the experimental data.

The TEM image selected in Figure 12 illustrates a few typical nanocubes of the sample nPSize04. Some particles are not completely formed and deviate strongly from the ideal cube, see Figure 12a. It should be noticed that other perfectly cubic particles are not correctly oriented on the substrate and can also negatively impact the measurements by microscopy (Figure 12b): when the (100) crystal plane is not in parallel contact with the sample surface, the determined dimensions will not correspond to the expected parameters. The amount of poorly formed particles has been estimated at around 25% and if we add the fraction of inclined cubes (i.e., the side not in full contact with the substrate), this rises to more than 40% of “imperfect” nanocubes.

Table 8 indicates the observed discrepancy between the characteristic size parameters measured on the perfect cubes compared with the total population including both the poorly formed and inclined fractions of NPs. The data show that this discrepancy falls within the measurement uncertainties, and it is not necessary to take this aspect separately into consideration. In fact, an ensemble measurement is comparable to a differentiated analysis not taking into account the poorly formed and inclined NPs.

To study in more detail the impact of poorly formed particles on the reliability of the size measurements for the gold nanocubes, an analysis with random draws was undertaken and the results are reported in Figure 13. First, visual particle classification was carried out according to their shape imperfections in the two classes: perfect nanocubes and imperfect ones (slightly larger). Then, several random draws were performed varying the proportion of each population. The error was then calculated by determining the deviation with an ideal population composed of perfect cubes. Considering that 40% of imaged objects are not perfectly cubic (i.e., 60% of the perfect cubes in Figure 12), this confirms that the error on the characteristic parameter measurements is roughly 1 nm.

### 4.4. Gold Nanorods

If we exclude the roundness at the edges, the rod-shaped nano-objects are sufficiently and accurately described by using two parameters, the length of the major axis, *L* and the diameter of their section, *D*. The measurands *D*_MinFeret_ and *D*_max.⟂_ are particularly well suited for determining these two parameters by EM-based techniques. The nanorod section can be directly measured by *h*_AFM_, but the *L* measurements are affected by the strong convolution between the AFM probe tip and the particles at the nanoscale. For this reason, only *h*_AFM_ for the particles section is determined by AFM. The results of the nPSize07 material obtained by all laboratories using microscopy (TEM, TSEM, SEM, and AFM) are reported in Figure 14. The error bars (k = 2) have been included, too. It can be noticed that the spread of data is fairly low apart from one AFM result. The observed discrepancies are very often within the error bars. The TEM image given in Figure 14 (right) shows one particle with a rather isotropic shape. Such irregularities represent roughly 6% of the entire population and definitively have no significant impact on measurement reliability. Furthermore, unlike the nanobipyramid measurement process, the nanorods have a single possible orientation on the substrate, as long as they are non-overlapped. The long axis of NPs tends to be parallel with the surface of the substrate, reducing the measurement errors in microscopy.

### 4.5. Outcome of the ILC

Table 9 gives the dispersion of the measurements for each nPSize material and each family of techniques. Overall, the data dispersion roughly corresponds to the mean uncertainties for the measurements of NPs with spherical shapes and even with cubic and elongated shapes. Regarding the nanocubes and nanorods, the height measurements carried out by AFM increased the dispersion due to issues with sample preparation and difficulties relative to a small number of counted NPs. The material that caused the most metrological challenges was, as expected, the population of bipyramids. Discrepancies of more than 15% were noted among the different techniques regarding the *s* and *L* results. In this case, the measurement uncertainties reached 10 nm. The different possible orientations of the particles on the substrate explain these deviations. Moreover, some very interesting results were obtained with the SAXS technique tested on the near-spherical nanoparticle populations, but also on particles with more complex shapes such as bipyramids, when the NP geometry is assumed in the model. Slight discrepancies are often observed between the measurements performed with EM and SAXS, especially for near-spherical particles. Here, the question can arise as to whether the SAXS results on the nanobipyramids are more reliable than EM ones, because of the independence of the SAXS signal of the NP orientation in the liquid suspension. For this type of complex shape, an approach could be to use SAXS data for reliably determining only the *s* parameter, and the *L* parameter would result from the assumption of a bipyramidal shape with the *s*/*L* ratio fixed from the information fed by microscopy. Hence, the two techniques would complement each other for reaching size measurements depending on the orientations of particles on the substrate.

## 5. Conclusions

The purpose of the nPSize project was to develop reference nanomaterials of more complex morphologies. This part of the standardization work is essential to improve the traceability and reliability of the nanoparticle size measurements. Bimodal populations as well as complex shaped nanoparticles (bipyramids, cubes, and rods) were synthesized and an ILC of measurements performed with different techniques (EM, AFM, and SAXS) was organized. The nanoparticle (NP) bimodal populations can be used as reference materials for accurately determining the NP concentration/size by SAXS or sp-ICP-MS (single-particle inductively coupled plasma mass spectrometry). Following the guidance given in this paper, nanoparticles with complex shapes can be suitable for calibrating the instruments involved in a hybrid metrology approach (for instance, the use of bipyramids by combining AFM and SEM) or for comparison of results obtained by AFM or SEM (nanocubes and nanorods). As a specific point, the gold nanocubes presented long-term stability issues, which must be better understood before further characterization.

Regarding ILC, first, the measurands defining the dimensions and the geometry of each type of particle were carefully defined. Furthermore, different strategies of measurement could be tested. Regarding the microscopy techniques (TEM, SEM, TSEM, and AFM), the reachable dimensional parameters depend on the physical principle of the method. Only the lateral size of the nano-objects is measurable by electron microscopy and only the height was measured by AFM, due to the convolution between the tip and the nanoparticle. The SAXS gave complementary and determining information, because of its ensemble character and, in our case, the result does not depend on the orientation of the nanoparticles. Actually, in the SAXS measurement of the suspended and isotropically aligned nanoparticles, the orientation-averaged scattering signal of the particles is measured. As expected, when all of the techniques are compared, the most consistent results with the lowest data dispersion were obtained with near-spherical nanoparticles, even if the population is bimodal. Nevertheless, the discrepancies among the results obtained with the different techniques are also very low and comparable to the corresponding method uncertainties for the cases of the nanocubes and nanorods. In contrast, the bipyramids gave rise to significant difficulties because the results of the size measurements obtained with microscopy techniques depend on the orientation of the particles on the substrate. An approach of hybrid metrology correlating SEM with AFM is proposed to noticeably improve the accuracy of the measurements of the parameters characterizing the geometry of such complexly shaped particles. A comparison with the SAXS results showed that a hybrid method approach correlating SAXS with electron microscopy is relevant as well. A quick analysis based on a few EM images indicating the NP geometry could be used as input to the SAXS model. In the case of bipyramids, the SAXS technique can benefit from providing data-independent NP orientation and providing a result representing a large volume of the sample.

## Figures and Tables

**Figure 1 nanomaterials-13-00993-f001:**
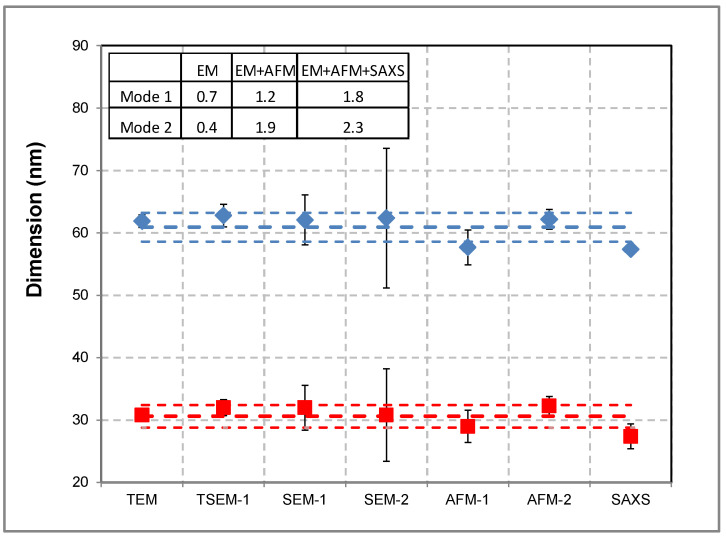
Size measurements of nPSize01 material (bimodal nanogold) performed with different techniques and by several laboratories. *D*_eq_ was determined by TEM, TSEM, and SEM, *h*_AFM_ was determined by AFM, and *D*_SAXS_ was determined by SAXS. Modes 1 and 2 are represented by red and blue symbols, respectively. The table in the inset gives the data spread calculated with standard deviations by taking into account the measurements by EM (electron microscopy) only, EM + AFM, and EM + AFM + SAXS. The uncertainty bar (k = 2) for some points is not visible when the associated uncertainty is too small. The dotted lines correspond to the mean value associated with the standard deviations calculated from the results obtained by all techniques.

**Figure 2 nanomaterials-13-00993-f002:**
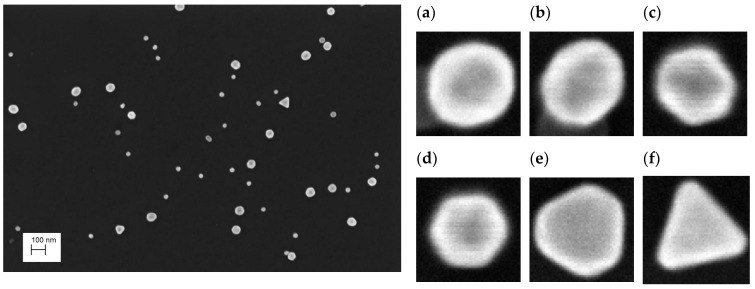
(**left**) Typical SEM image of the nPSize01 gold NPs and (**right**) various nanoparticle shapes (**a**–**f**) found in the sample: some examples are given of near-spherical shapes (**a**,**b**), icosahedral and decahedral shapes (**c**,**d**) and truncated tetrahedral shape (**e**,**f**).

**Figure 3 nanomaterials-13-00993-f003:**
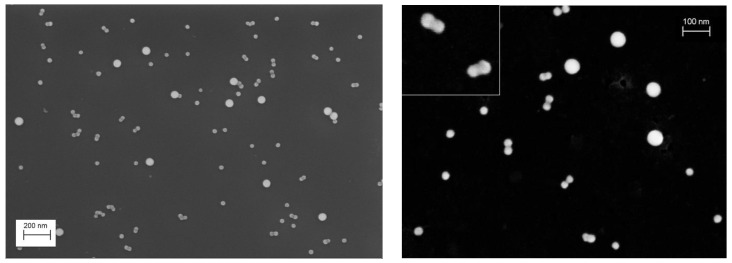
Representative SEM images of nPSize12 NPs (bimodal SiO_2_).

**Figure 4 nanomaterials-13-00993-f004:**
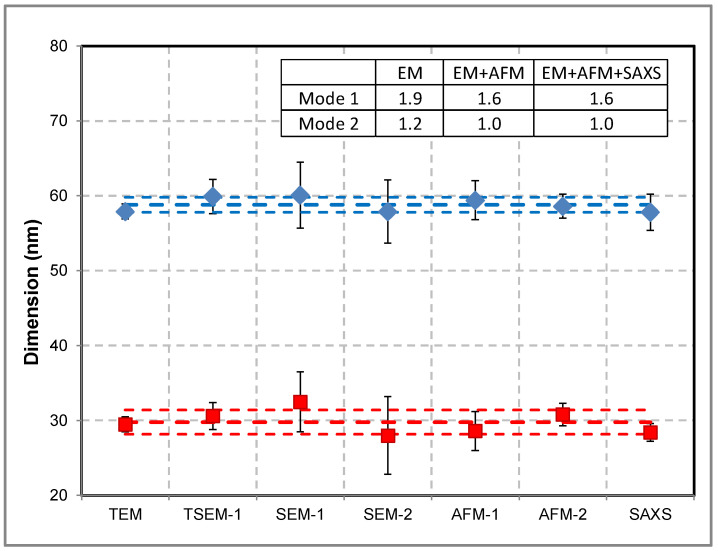
Size measurements of nPSize12 material (bimodal nanosilica) performed with different techniques and by several laboratories. *D*_eq_ was determined by TEM, TSEM, and SEM, *h*_AFM_ was determined by AFM, and *D*_SAXS_ was determined by SAXS. Modes 1 and 2 are represented by red and blue symbols, respectively. The inset table gives the data spread calculated with the standard deviation by taking into account the measurements with EM (electron microscopy), EM + AFM, and EM + AFM + SAXS. The uncertainty bar (k = 2) is not visible when the associated uncertainty is too small. The dotted lines correspond to the mean values associated with standard deviations calculated from the results obtained by all techniques.

**Figure 5 nanomaterials-13-00993-f005:**
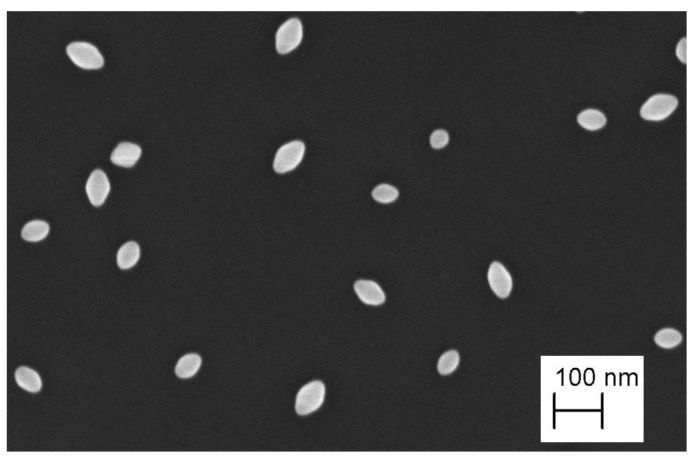
SEM image of nanoscale bipyramidal particles (nPSize03).

**Figure 6 nanomaterials-13-00993-f006:**
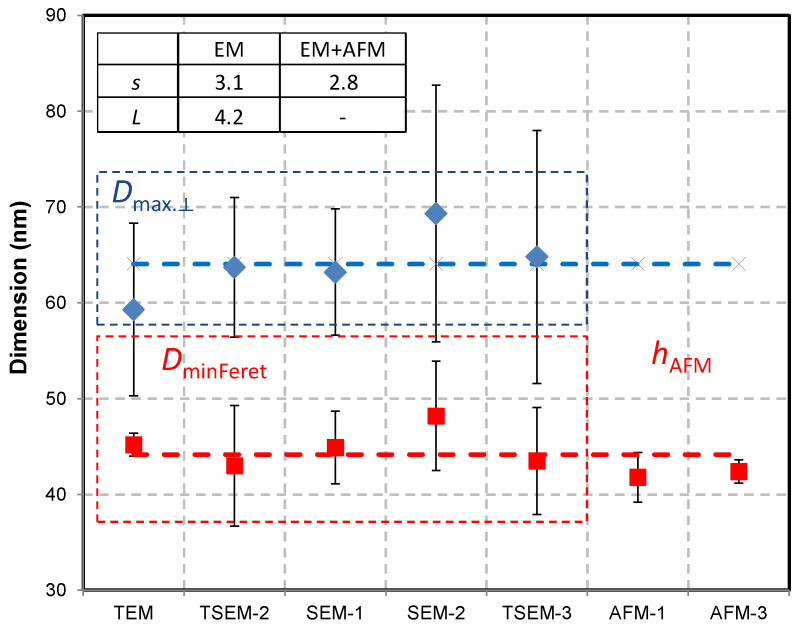
Size measurements of the nPSize03 material (nanoscale bipyramids) performed with different techniques and several laboratories. With EM, the measurands *D*_MinFeret_ and *D*_max.⟂_ allow the parameters *s* and *L* to be measured, such that the side of the square base and the length of the major axis are assessed, respectively. The inset table gives the data spread calculated with the standard deviation by taking into account the measurements with EM (electron microscopy) and EM + AFM. The uncertainty bars (k = 2) are also given with each laboratory mean value (lognormal distribution).

**Figure 7 nanomaterials-13-00993-f007:**
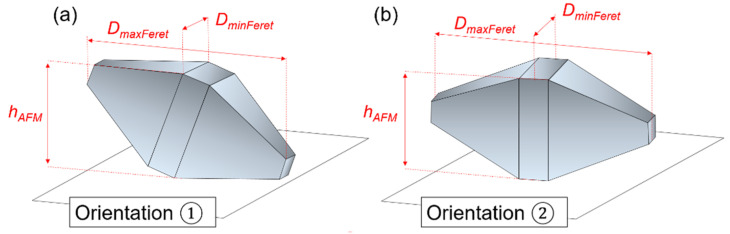
Schematic view of the two possible orientations, noted 1 (**a**) and 2 (**b**), of nanobipyramides on the substrate after deposition for microscopy measurements.

**Figure 8 nanomaterials-13-00993-f008:**
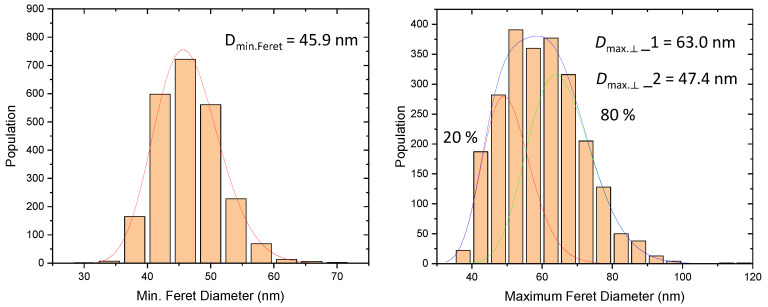
(**Top**) Histograms of size distribution linked to the *D*_MinFeret_ and *D*_max.⟂_ measurements. (**Bottom**) Histogram of the aspect ratio of the nPSize03 nanoparticles with representative SEM and AFM images for a few particle classes.

**Figure 9 nanomaterials-13-00993-f009:**
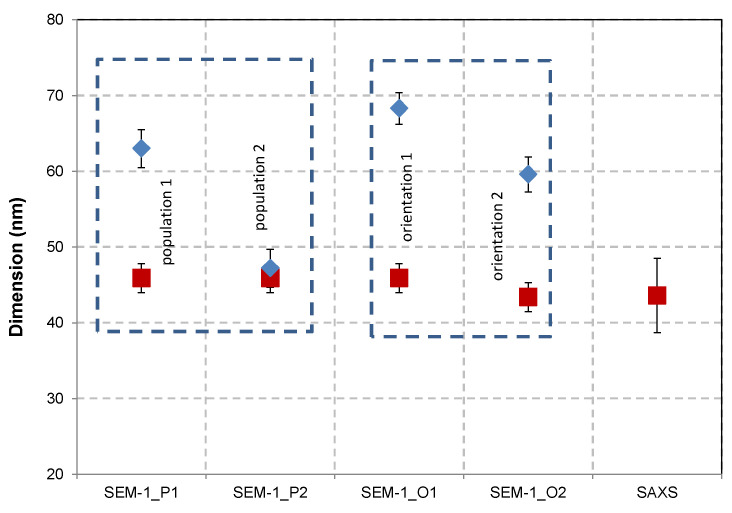
Parameters s (red) and L (blue) measured by SEM (*D*_MaxFeret_ and *D*_MinFeret_) as a function of their aspect ratio (left box, 2000 NPs, population P1 and P2) and their orientation (middle right box, 250 NPs, orientations O1 and O2). The SAXS results (*s*_SAXS_) are added for comparison; a *L*_SAXS_ value was calculated by using the aspect ratio deduced from microscopy data. The uncertainty bars (k = 2) are also given together with the mean value.

**Figure 10 nanomaterials-13-00993-f010:**
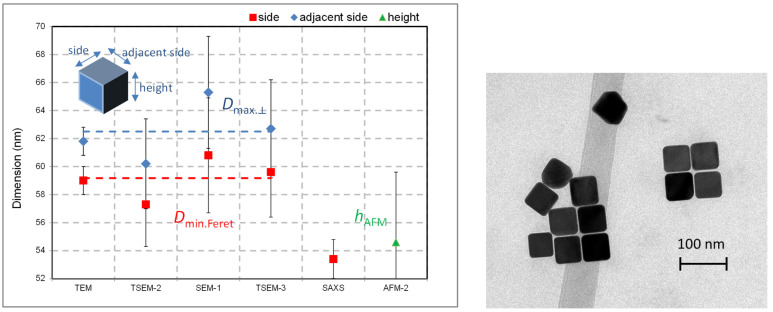
(**Left**) Size measurements of nPSize04 material (nanocubes) performed by different techniques and laboratories. (**Right**) TEM image of Au nanocubes. The uncertainty bars (k = 2) are also given with the mean values.

**Figure 11 nanomaterials-13-00993-f011:**
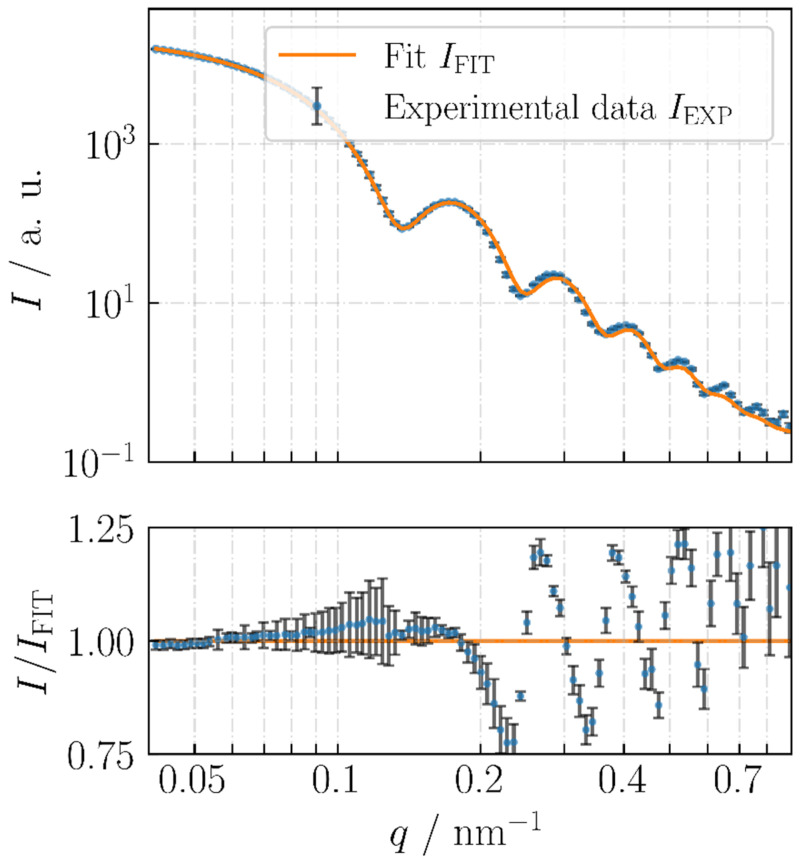
Experimental SAXS curve (symbols) of the nPSize04 (nanocubes) material together with the fit for cubes with rounded edges (solid orange line).

**Figure 12 nanomaterials-13-00993-f012:**
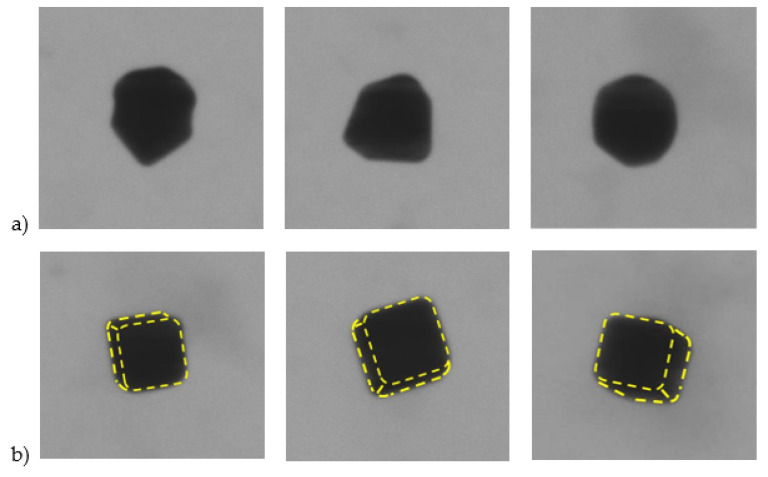
Examples of (**a**) non-cubic particles or (**b**) inclined nanocubes (side not parallel with the substrate).

**Figure 13 nanomaterials-13-00993-f013:**
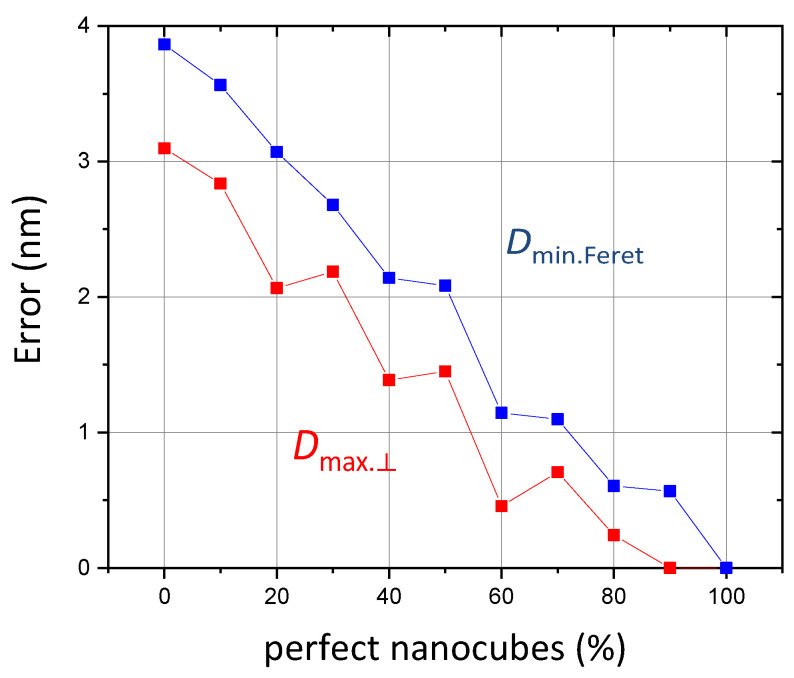
Expected error calculated by random draw analysis for different fractions of perfect/imperfect nanocubes.

**Figure 14 nanomaterials-13-00993-f014:**
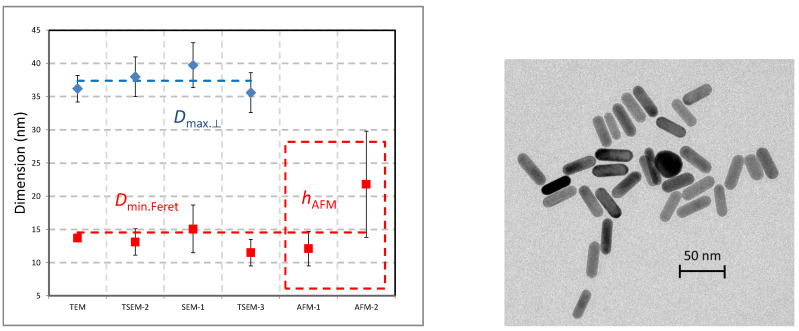
(**Left**) Size measurements of the nPSize07 material (Au nanorods) performed by different techniques and laboratories. (**Right**) Representative gold nanorods imaged by TEM. The uncertainty bars (k = 2) are also given with the mean values.

**Table 1 nanomaterials-13-00993-t001:** Information on the materials investigated in this study including the chemical composition, shape, mode number, and geometry.

	Simple Shape Isotropic		Complex Shape Anisotropic		
Sample Name	nPSize01	nPSize12	nPSize03	nPSize04	nPSize07
Description	Bimodal gold	Bimodal silica	Bipyramidal titania	Gold nanocubes	Gold nanorods
Shape	Spheroid	Sphere	Bipyramid	Cube	Rod
Number of modes	2	2	1	1	1
Nominal size	30 nm/60 nm	30 nm/60 nm	45 nm/60 nm	60 nm	15 nm/40 nm
Polydispersity	Shape/size	Size	-	-	-
Expected truncation *	Yes	No	Yes	Yes	No

* “expected truncation” means possible deviation from the ideal shape. **nPSize01/bimodal gold:** Citrate-stabilized colloidal gold NP suspensions with a diameter of approximately 30 nm and 60 nm and dispersed in water were purchased from a commercial supplier, BBI Solutions Ltd. (Nottingham, UK). The two suspensions were further processed as received by mixing at a ∼1:1 ratio (particle number concentration based), ampouled using 5 mL amber glass vials, and packaged at LGC (Luckenwalde, Germany). The vials were sterilized using Co^60^ gamma-irradiation with a minimum dose of 35 kGy. Homogeneity and stability in terms of the number concentration of the two size fractions were tested with spICP-MS (frequency method) by LGC (Teddington, UK) to ensure that the material meets the set requirements and remains stable throughout its shelf-life. The value of the number concentration of the colloid gold particles was determined by LGC (Teddington, UK) with spICP-MS (frequency method) [15] and was (1.88 ± 0.24) × 10^13^ kg^−1^ and (1.93 ± 0.24) × 10^13^ kg^−1^ for the 30 nm and 60 nm fractions, respectively. **nPSize12/bimodal silica**: Colloidal silica NPs with nominal particle diameters of either 30 nm or 60 nm were synthesized by CEA with the PSD (particle size distribution) varying from <10% to ≈20%, following a well-known approach [16]. nPSize12 was obtained by gently mixing 30 nm and 60 nm suspensions in a 1:1 ratio (particle-number-based concentration). The concentration was measured using the CEA laboratory SAXS instrument (Xeuss 2) and protocol described in [17]. **nPSize03/bipyramidal titania: TiO_2_** anatase bipyramids with a high percentage of exposed {101} facets were synthesized by hydrothermal treatment of Ti(IV)-triethanolamine complex aqueous solution at 493 K, as described in detail in the literature [18,19]. **nPSize04/Gold nanocubes:** A suspension of mono-crystalline colloidal gold cubic NPs was synthesized by CEA following the well-known method utilizing cetyltrimethylammonium bromide (CTAB), as described elsewhere [20]. The material was kept as a single master batch and stored at (5 ± 4) °C. **nPSize07/Gold nanorods:** A suspension of monocrystalline colloidal gold nanorods was synthesized by CEA following the classical protocol of the seeding growth method described elsewhere [21]. The gold nanorods were purified by tree washing stages from centrifugation. The suspension was stored at 8 °C.

**Table 2 nanomaterials-13-00993-t002:** Sources of the uncertainty budget for NP size measurements by TEM.

Source	Contribution
Sample preparation	Significant (<3 nm)
Repeatability	Medium for complex sample (<1.5 nm) Minor for spherical and monomodal populations (<1 nm)
Statistics	Minor (<1 nm)
Thresholding	Minor (<1 nm)
Boundary determination	Minor (<1 nm)
Pixel size	Minor (<0.5 nm)
Calibration	Minor (<0.4 nm)

**Table 3 nanomaterials-13-00993-t003:** Main uncertainty sources for NP size measurements by SEM.

Source	Contribution
e-beam size	Significant (<2 nm)
Sample preparation	Significant (depending on the sample)
Thresholding	Significant (<2 nm)
Repeatability	Medium (<0.5 nm)
Magnification/pixel size	Medium (<0.5 nm)
Contamination	Medium (<1–2 nm)
Beam damage	Minor (when taking precautions)
Orientation/adhesion on the surface	Minor (if near-spherical NPs)

**Table 4 nanomaterials-13-00993-t004:** Main uncertainty sources for NP size measurements by TSEM.

Source	Contribution
Thresholding	Significant (1–7 nm)
Sample preparation	Significant (depending on the sample)
Selection of particles and statistics	Medium (<1.5 nm)
Effects of TSEM imaging	Medium (<1 nm)
Determination of pixel size	Minor (<0.2 nm)

**Table 5 nanomaterials-13-00993-t005:** Sources of uncertainty for NP size measurements by AFM (k = 1).

Source	Contribution
Repeatability	Significant (≈1 nm)
Amplitude set point *	Significant (≈1 nm)
Calibration *	Medium (<1 nm)
Operator *	Medium (<1 nm)
Scan speed *	Medium (<1 nm)
Temperature drift *	Medium (<1 nm)
Image analysis *	Medium (<1 nm)
Baseline roughness	Medium (<1 nm)
Resolution limit along the Z-axis	Minor (<0.1 nm)
XY contributors (pixel size, resolution limit noise along XY-axis)	Minor (<0.1 nm)

* a qualified operator should be able to properly calibrate the instrument and optimize the settings and conditions to minimize the uncertainties for these parameters. Some of the values are based on worst-case fluctuations for extreme settings. Furthermore, this budget can change depending on the model of the instrument.

**Table 6 nanomaterials-13-00993-t006:** Main uncertainty sources for NP size measurements by SAXS.

Source	Contribution
Model fitting	Significant (a few % depending on the sample polydispersity)
Detector pixel size	Minor (10^−3^)
Distance sample–detector	Minor (2. × 10^−4^)
Photon energy	Minor (10^−4^)
Beam center	Minor (10^−4^)

**Table 7 nanomaterials-13-00993-t007:** Parameters describing the NP geometry for each NP material and the measurands.

Sample	Shape	Parameters	Measurands
EM	AFM	SAXS
nPSize01	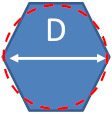	*D* = equivalent circular diameter (ECD)	*D* _eq_	*h* _AFM_	*D* _SAXS_
nPSize12	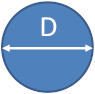	*D* = sphere diameter	*D* _eq_	*h* _AFM_	*D* _SAXS_
nPSize03	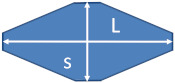	*L* = length (major axis)*s* = side of the square base	*D*_MinFeret_,*D*_MaxFeret_ *	*h* _AFM_	*s* _SAXS_
nPSize04	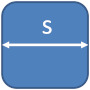	*s* = side	*D*_MinFeret_,*D*_MaxFeret_ *	*h* _AFM_	*s* _SAXS_
nPSize07	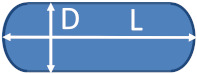	*L* = length*D* = section diameter	*D*_MinFeret_,*D*_MaxFeret_ *	*h* _AFM_	-

(*) the maximum Feret diameter is determined in this study as the distance between two parallel tangential lines in the direction perpendicular to the *D*_MinFeret_.

**Table 8 nanomaterials-13-00993-t008:** SEM size measurements of the perfect nanocubes compared with the total population including the “badly” oriented and poorly formed particles.

Mesurand	Perfect Nanocubes	Total Population
Minimum Feret diameter	(59.9 ± 2.0) nm	(60.8 ± 2.1) nm
Maximum Feret diameter	(63.3 ± 1.9) nm	(65.3 ± 1.9) nm

**Table 9 nanomaterials-13-00993-t009:** Dispersion of the data measured by the different techniques used in the nPSize project as a function of the particle population.

Type of the Nanoparticulate Material		EM			EM + AFM	EM + AFM + SAXS
		*D* _eq_	*D* _Min_ _Feret_	*D* _MaxFeret_	*D* _eq/_ *h* _AFM_ *D* _MinFeret/_ *h* _AFM_	*D*_MinFeret/_*D*_SAXS_ *D*_eq/_*D*_SAXS_/*h*_AFM_
Spherical shape	Bimodal population	<1 nm	-	-	<2 nm	≈2 nm
Complex shape	Bipyramids	-	≈2.5 nm	≈5 nm (depends on the orientation)	≈2.5 nm (for *s*)	≈2.5 nm (for *s*) and ≈5 nm (for *L*)
	Nanocubes	-	<2 nm	< 2 nm	<2.5 nm	-
	Nanorods	-	<2 nm	≈2 nm	≈4 nm	-

## Data Availability

Not applicable.

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
