# Peer review of "Metrological Protocols for Reaching Reliable and SI-Traceable Size Results for Multi-Modal and Complexly Shaped Reference Nanoparticles"

_nanomaterials, 2023, doi:10.3390/nano13060993_

Round 1

Reviewer 1 Report

This article explored different nanoparticle size measurement techniques on reference nanomaterials, including TEM, SEM, TSEM, AFM and SAXS. The measurement results were compared and insights are gained from this study about the measurement techniques. It has been a challenge in the field of nanoparticles to accurately characterize the particle size, especially when the shape of these nanoparticles are anisotropic. Furthermore, different methods are usually used in literature making a direct nanoparticle size comparison difficult between studies. Some form of calibration/comparison of different characterization methods is welcomed to address this issue. The insight from this study comparing different measurement techniques and relating them to the physical properties of the samples provided a valuable reference for future nanomaterial studies. The idea for this standardization work is clear and the data is helpful because it included nanomaterials with multiple non-spherical shapes. I recommend acceptance in the current form.

Author Response

Thank you for your comments.

Reviewer 2 Report

1) In part the work recapitulates previous results in the nPSize project. The substantially new part is the inter-laboratory comparison. The authors demonstrated the ability to achieve consistent results among the different laboratories and deal with various ambiguities and discrepancies. And this is the perspective emphasised in the conclusion. For some materials, however, the characterisation required more effort, more complex steps or consideration of more subtle points. The declared goal is to propose new reference nanoparticle materials. Hence, it is crucial that everyone will be able to use them reliably as a reference (for some reasonable value of ‘everyone’). Some nanomaterials seem generally better or worse suited than others, some may be better or worse for specific purposes. Such critical evaluation of the different materials is missing and needs to be added – where else it should be done if not here?

2) The authors made little effort to put the work into context and cite any relevant literature beside their own papers. Not counting ISO standards, the self-citation ratio is about 80%.

3) Reference number 01 on l. 210 should be probably 10.

4) I suggest transposing table 1. It would make it easier to fit – and permit adding more overview parameters the reader now struggles to gather, such as nominal particle sizes. The meaning of ‘expected truncation’ column is unclear.

5) What software was used to process AFM-3 and AFM-3 data – Platypus? The text is only clear about AFM-1.

6) The nonstandard ‘maximum Feret diameter’ does not measure the maximum of anything. A different term and symbol have to be chosen. Since it measures the dimension in an orthogonal direction, it can for instance be called orthogonal and ⟂ used in subscript/superscript.

7) The entire section 4.1 describes methodology and does not belong to results.

8) Figure 1 and following similar tables list ‘data spreads’. Which quantity characterising data spread the numbers correspond to?

9) Deviations of individual instruments from the overall mean in figures 1, 4, 6, etc. seem correlated. Meaning individual instruments seem to provide systematically lower or higher values compared to the overall mean (although the later figures are more difficult to interpret). Has any cross-sample analysis been done? Have any systematic deviations of instruments/techniques been identified?

10) Clarify ‘Even if the agglomeration state on the substrate is as reduced as for the sample nPSize01…’ (l. 490). As I understand, the agglomeration was not actually reduced comparably, but the same measures to reduce were taken.

11) The text states (l. 522) ‘These NPs are (anatase) truncated tetragonal bipyramids with height equivalent {101} faces and four {100} faces on the edge of square base.’ Beside only mentioning 12 of the 14 faces, there seems to be a typo: height in place of eight. However, ref. 20 contains exactly the same statement about height equivalent faces. This reviewer is perplexed.

12) Were the AFM results for bipyramids filtered (how?) to only include orientation â‘¡? Or do the results in figure 6 also include the slightly longer height for orientation â‘ ?

13) How can the mode of DMaxFeret be close to the DMinFeret histogram (l. 558)?

14) Text around l. 580 about ensemble and counting methods repeats the introduction.

15) What did the Monte Carlo simulation (l. 653) simulate? The entire imaging process or just the mixing statistics? If the former, what were the shapes of imperfect cubes – according to figure 13 and table 8 they seem to be larger, but that is the only clear information. If the latter then would not a simple analytic computation give the plot in figure 13?

16) The language should be checked. There are numerous various odd formulations and/or incorrect grammar (for each sizing methods; distributions of the minimum Feret diameter, maximum Feret diameters; impact on measuring results; rises to more 40% ‘imperfect’ nanocubes; shape … is quite various and sometimes far from a sphere; etc.)

Author Response

Comments and Suggestions for Authors

  • In part the work recapitulates previous results in the nPSize project. The substantially new part is the inter-laboratory comparison. The authors demonstrated the ability to achieve consistent results among the different laboratories and deal with various ambiguities and discrepancies. And this is the perspective emphasised in the conclusion. For some materials, however, the characterisation required more effort, more complex steps or consideration of more subtle points. The declared goal is to propose new reference nanoparticle materials. Hence, it is crucial that everyone will be able to use them reliably as a reference (for some reasonable value of ‘everyone’). Some nanomaterials seem generally better or worse suited than others, some may be better or worse for specific purposes. Such critical evaluation of the different materials is missing and needs to be added – where else it should be done if not here?

In the nPSize project a well-selected set of 12 types of reference nanoparticles have been metrologically characterised:

  1. 3 materials with accurate concentration: 2 bimodal gold NPs in 2 different relative concentrations and bipyramidal titania,
  2. 3 materials with a non-spherical shape: gold cubes, titania platelets and acicular titania, additionally to the bipyramidal titania from point i), and
  • 5 materials with different polydispersity of the particle size: non-monodisperse Au nanorods, monomodal silica with narrow PSD, monomodal silica with a broader PSD, 2 bimodal silica Ps in 2 different relative concentrations.

Of these 12 nPSize materials, 5 materials are presented in the current paper with detailed characterisation including assessment of their suitability as certified reference materials. Various reasons have contributed to the decision to disqualify remaining 3 nPSize materials: a too broad size distribution coupled with a shape which deviates significantly from the ideal square or rod-shape (respectively with hardly definable nano-descriptors for the acicular titania), and an unattained PSD of defined larger width for the silica monodisperse nanoparticles. It should be noted that these ‘imperfections’ in the material synthesis could eventually be supressed, however, much more time and effort are needed to be invested. Alternatively, more advanced characterisation approaches could solve the consideration of such size and shape imperfections, which bring our reference nanoparticles closer to real commercial nanomaterials. Machine learning approaches fed with accurate analysis from manual training could be another way of progressing with such complex types of nanoparticles. Related work in different laboratories worldwide is in progress. Further, it should be noticed that, e.g. for nanoplatelets there is still no standard procedures to measure accurately and representatively the thickness. The best way is probably the development of a sample preparation protocol which guarantees (a majority) of single platelets on a substrate. A method combination of SEM for lateral size and AFM for the thickness appears to be the only solution for the complete and accurate geometrical characterisation of nanoplatelets. The current state is that such a sample preparation procedure could not be reached yet. Here, too, machine learning approaches might be more successful, but need first to be developed and then properly tested.

Of the 5 nanoparticle materials presented in this paper the gold nanocubes have presented long-term stability issues, which must be further understood before further characterisation. The bimodal gold, bimodal silica, bipyramidal titania and gold nanorods are suited candidates to be further developed as certified reference materials. While the bimodal gold and silica nanoparticles can be prepared in larger amounts as certified reference materials, the gold nanocubes and nanorods could be produced at a reduced, laboratory scale. The same is the case for the bipyramidal titania. To note that the latter material (nPSize03 in this paper) and the silica bimodal materials (here with nPSize12 as one of them) are part of two interlaboratory comparisons under VAMAS as Projects #15 and #16 under TWA 34 ‘Nanoparticle populations’ (http://www.vamas.org/twa34/documents/2021_vamas_twa34_p15_tio2_bp_np.pdf and http://www.vamas.org/twa34/documents/2021_vamas_twa34_p16_sio2_npbimos.pdf). More than 100 ampoules of each are available for free at BAM (contact Dan Hodoroaba).

Some considerations and information about potential CRM were added in the conclusion.

  • The authors made little effort to put the work into context and cite any relevant literature beside their own papers. Not counting ISO standards, the self-citation ratio is about 80%.

We added some references about comparison studies conducted in a recent past and performed by other authors. We limited also self-referencing.

  • Reference number 01 on l. 210 should be probably 10.

The correction has been done.

  • I suggest transposing table 1. It would make it easier to fit – and permit adding more overview parameters the reader now struggles to gather, such as nominal particle sizes. The meaning of ‘expected truncation’ column is unclear.

We have transposed the table and added a row relative to nominal particle sizes. “Expected truncation” means that we can expect non ideal shape.

  • What software was used to process AFM-3 and AFM-3 data – Platypus? The text is only clear about AFM-1.

This point was clarified in the text in a separate part at the end of the section.

  • The nonstandard ‘maximum Feret diameter’ does not measure the maximum of anything. A different term and symbol have to be chosen. Since it measures the dimension in an orthogonal direction, it can for instance be called orthogonal and ⟂ used in subscript/superscript.

In text and Figures, Dmax.Feret was replaced by Dmax.⟂

  • The entire section 4.1 describes methodology and does not belong to results.

That’s right. This part was removed from the result section and integrated in section 3.7.

  • Figure 1 and following similar tables list ‘data spreads’. Which quantity characterising data spread the numbers correspond to?

The data spread was calculated with standard deviation from measurements given by laboratories by taking into account some groups of techniques. This explanation was added in the captures.

  • Deviations of individual instruments from the overall mean in figures 1, 4, 6, etc. seem correlated. Meaning individual instruments seem to provide systematically lower or higher values compared to the overall mean (although the later figures are more difficult to interpret). Has any cross-sample analysis been done? Have any systematic deviations of instruments/techniques been identified?

You are right for some laboratories / instruments (TSEM-1, SEM-1, AFM-2 and SAXS), but it is not systematically. Furthermore, in all cases (except for one), the uncertainties cover the overall mean. That means the systematic errors was correctly identified in the uncertainty budget.

  • Clarify ‘Even if the agglomeration state on the substrate is as reduced as for the sample nPSize01…’ (l. 490). As I understand, the agglomeration was not actually reduced comparably, but the same measures to reduce were taken.

What we meant was the dispersion state of the two samples (nPSize01 and nPSize12) is similar but the presence of dimers and trimers is observed in nPSize12. Maybe, the word “agglomeration” is not relevant here. We reworded with dispersion more suitable.

  • The text states (l. 522) ‘These NPs are (anatase) truncated tetragonal bipyramids with height equivalent {101} faces and four {100} faces on the edge of square base.’ Beside only mentioning 12 of the 14 faces, there seems to be a typo: height in place of eight. However, ref. 20 contains exactly the same statement about height equivalent faces. This reviewer is perplexed.

You are right. “Height” was replaced by “eight”. The two forgotten faces were added in the text.

  • Were the AFM results for bipyramids filtered (how?) to only include orientation â‘¡? Or do the results in figure 6 also include the slightly longer height for orientation â‘ ?

Actually, the orientations â‘  and â‘¡ can be easily discriminated by AFM because the upper faces are clearly visible. If you look at the Figure 7, in the case â‘ , the upper face is {101} and {100} in the case â‘¡. This part is described in details in the ref. [L. Crouzier et al., Nanomaterials 2021, 11, 3359]. But, regarding the AFM measurements reported in Figure 6, your observation is correct, all orientations are included. The observed discrepancy with other techniques can be explained by the fact that the slightly longer height are also measured.

  • How can the mode of DMaxFeret be close to the DMinFeret histogram (l. 558)?

In fact, the nano-bipyramids consist of two populations. The first one is composed of larger and completely-formed particles with an aspect-ratio very close to 0.75. Regarding the second population, the particles are smallest with a quasi-spherical shape. Consequently, their AR is roughly 1 and the Feret diameters are similar. The sentence (l. 558) is related to this second population of smaller particles.

  • Text around l. 580 about ensemble and counting methods repeats the introduction.

Exact, the sentence was reduced.

  • What did the Monte Carlo simulation (l. 653) simulate? The entire imaging process or just the mixing statistics? If the former, what were the shapes of imperfect cubes – according to figure 13 and table 8 they seem to be larger, but that is the only clear information. If the latter then would not a simple analytic computation give the plot in figure 13?

You are right, there is a big mistake. It is not a MonteCarlo simulation but a random draw analysis, so just mixing statistics. This part was changed in the text. First, a visual particle classification was carried out according to the NP shape imperfections in two classes: perfect nanocubes or not. Then, several random draws were performed varying the proportion of each population. Error is then calculated by determining the deviation with an ideal population composed of perfect cubes.

  • The language should be checked. There are numerous various odd formulations and/or incorrect grammar (for each sizing methods; distributions of the minimum Feret diameter, maximum Feret diameters; impact on measuring results; rises to more 40% ‘imperfect’ nanocubes; shape … is quite various and sometimes far from a sphere; etc.)

Regarding the Feret diameters. According ISO 21363, the definitions are :

3.4.3 Feret diameter

distance between two parallel tangents on opposite sides of the image of a particle (3.1.3)

[SOURCE: ISO 13322-1:2014, 3.1.5, modified — Note 1 to entry has been added.]

3.4.4 maximum Feret diameter

maximum length of an object whatever its orientation

[SOURCE: ISO/TR 945-2:2011, 2.1, modified — Note 1 to entry has been deleted.]

3.4.5 minimum Feret diameter

minimum length of an object whatever its orientation

Otherwise, the language has been checked, mistakes removed, and formulations improved.

Reviewer 3 Report

The manuscript rather presents a technical report than a scientific paper. Not the entire audience of the journal is familiar with the nPSize project, its goals and basics, and reference to it throughout the paper does not add more significance to the research presented. However, the aim to compare the performance of different methods that can be applied for measurements at the nanoscale is noteworthy and the manuscript contains a bunch of methodological information that can be of use for the audience of Nanomaterials.

However to be eligible for publication the paper needs corrections. My major concerns are as follows.

1.      The title is too pretentious. In fact, the paper presents the comparison of performance and accuracy of a set of methods used for measurements of sizes of NPs of different (regular) shapes.

2.      The Introduction section does not contain any discussion of numerous measurements of NP sizes performed by other authors, including the analysis of their measurement errors. The substantial discussion should be added, along with corresponding references. This will also help to avoid excessive self-referencing (14 of 23 refs).

3.      The correct performance of the software utilized in image analysis is crucial. The data obtained using the home-made soft cannot be considered reliable until it is extensively tested. If possible I would recommend to make the code freely available in the supplement, to allow TSEM users to test it.

There are also some minor corrections needed:

1.      References to tables and figures throughout the text are missing.

2.      Figures 8 and 10 need improvement, the labels are blurring.

3.      I don’t understand the need for two images in Fig. 5 with scale difference of only about 1.5.

4.      The size of capillaries used in SAXS experiments need to be indicated.

5.      The description of TEM technique in sec. 2.2.1 is too detailed and can be shortened.

6.      At the same time the procedures used for NP fabrication are described to concisely and references are made to other papers, that makes difficulties for readers. I would suggest to describe these procedures in more detail and to move these data to supplementary material.

Author Response

Comments and Suggestions for Authors

The manuscript rather presents a technical report than a scientific paper. Not the entire audience of the journal is familiar with the nPSize project, its goals and basics, and reference to it throughout the paper does not add more significance to the research presented. However, the aim to compare the performance of different methods that can be applied for measurements at the nanoscale is noteworthy and the manuscript contains a bunch of methodological information that can be of use for the audience of Nanomaterials.

Further information about nPSize project was added in the introduction.

Moreover, several interlaboratory comparison exercises conducted in a recent past were detailed in the introduction. Most of published papers deal with studies on spherical, monodispersed and well dispersible nanoparticles with a single or multi-techniques. We show the interest of our work using complex-shaped or multimodal nanoparticles with a very good consistency of results between the used characterization methods.

However to be eligible for publication the paper needs corrections. My major concerns are as follows.

  1. The title is too pretentious. In fact, the paper presents the comparison of performance and accuracy of a set of methods used for measurements of sizes of NPs of different (regular) shapes.

We have changed the title to make it less pretentious : “Metrological protocols for reaching reliable and SI-traceable size results for multi-modal and complex-shaped reference nanoparticles”

  1. The Introduction section does not contain any discussion of numerous measurements of NP sizes performed by other authors, including the analysis of their measurement errors. The substantial discussion should be added, along with corresponding references. This will also help to avoid excessive self-referencing (14 of 23 refs).

As mentioned above, we added some references about comparison studies performed by other authors and limited self-referencing.

  1. The correct performance of the software utilized in image analysis is crucial. The data obtained using the home-made soft cannot be considered reliable until it is extensively tested. If possible I would recommend to make the code freely available in the supplement, to allow TSEM users to test it.

You are right the software should be intensively used to test its reliability, but unfortunately, making the code freely available is impossible. Pollen Metrology (software developer) is a private company (French SME), with business objectives. Some sections of the nPSize signed documents provided the legal protection of intellectual properties.

There are also some minor corrections needed:

  1. References to tables and figures throughout the text are missing.

In the new version, references are visible.

  1. Figures 8 and 10 need improvement, the labels are blurring.

The Figures 8 and 10 have been modified and fonts increased. All details seems to be fine in the new version, clearly readable at high resolution at a monitor, also zoomable.

  1. I don’t understand the need for two images in Fig. 5 with scale difference of only about 1.5.

One of the two has been removed.

  1. The size of capillaries used in SAXS experiments need to be indicated.

Information added in the text in section 2.2.5.

  1. The description of TEM technique in sec. 2.2.1 is too detailed and can be shortened.

Description part on TEM was shortened.

  1. At the same time the procedures used for NP fabrication are described to concisely and references are made to other papers, that makes difficulties for readers. I would suggest to describe these procedures in more detail and to move these data to supplementary material.

A supplementary material with all sample preparation protocols was prepared.

Reviewer 4 Report

The paper discusses the size and shape information obtained independently in several laboratories from electron microscopy, atomic force microscopy, and small-angle X-ray scattering for a variety of nanoparticles (spherical and non-spherical), some with bimodal size distributions and others with monodisperse and evaluates the reliability of the measurements, including their differences.

Although individual researchers have traditionally been concerned about the differences in size and shape information that depend on various measurement methods and have discussed them separately, they are often biased toward microscopy or only using scattering methods. These methods must be complementary, and it is essential to identify their problems and understand the strengths and weaknesses of the measurements. It is worthwhile to summarize them straightforwardly.

The following should be corrected or added

In section 4.4 on gold nanoparticles, SAX data are presented in Figure 11, showing the experimental data, calculated results, and the degree of discrepancy between them. However, there is no comment on them and the discussion proceeds. What are the authors' points of contention about these?

In the conclusion section, it was stated that the results obtained by SAXS are independent of the orientation of the nanoparticles, but to be precise, that is incorrect. In the non-spherical case, in solution (dispersant), the nanoparticles must be randomly oriented, i.e., the nanoparticles (non-spherical) must be oriented in all directions relative to the incident X-rays.

For example, if the rod particles exist with their long axes oriented in a certain fixed direction relative to the incident X-ray beam, then the model calculations will consider their orientation, and the scattering results will differ from the random orientation. Please correct the wording to avoid misunderstanding.

Author Response

Comments and Suggestions for Authors

The paper discusses the size and shape information obtained independently in several laboratories from electron microscopy, atomic force microscopy, and small-angle X-ray scattering for a variety of nanoparticles (spherical and non-spherical), some with bimodal size distributions and others with monodisperse and evaluates the reliability of the measurements, including their differences.

Although individual researchers have traditionally been concerned about the differences in size and shape information that depend on various measurement methods and have discussed them separately, they are often biased toward microscopy or only using scattering methods. These methods must be complementary, and it is essential to identify their problems and understand the strengths and weaknesses of the measurements. It is worthwhile to summarize them straightforwardly.

The following should be corrected or added

In section 4.4 on gold nanoparticles, SAX data are presented in Figure 11, showing the experimental data, calculated results, and the degree of discrepancy between them. However, there is no comment on them and the discussion proceeds. What are the authors' points of contention about these?

A comment has been added in the text just below the Figure.

In the conclusion section, it was stated that the results obtained by SAXS are independent of the orientation of the nanoparticles, but to be precise, that is incorrect. In the non-spherical case, in solution (dispersant), the nanoparticles must be randomly oriented, i.e., the nanoparticles (non-spherical) must be oriented in all directions relative to the incident X-rays.

For example, if the rod particles exist with their long axes oriented in a certain fixed direction relative to the incident X-ray beam, then the model calculations will consider their orientation, and the scattering results will differ from the random orientation. Please correct the wording to avoid misunderstanding.

That is correct. One sentence was added in the conclusion to clarify.

Round 2

Reviewer 3 Report

The authors addressed all my concerns and the paper can be published in the current version.

Author Response

English was thoroughly revised.